# Patient–Practitioner–Environment Synchronization: Four-Step Process for Integrating Interprofessional and Distinctive Competencies in Osteopathic Practice—A Scoping Review with Integrative Hypothesis

**DOI:** 10.3390/healthcare13070820

**Published:** 2025-04-04

**Authors:** Christian Lunghi, Francesca Baroni, Giandomenico D’Alessandro, Giacomo Consorti, Marco Tramontano, Laurent Stubbe, Josie Conte, Torsten Liem, Rafael Zegarra-Parodi

**Affiliations:** 1BMS Formation, 75116 Paris, France; christian@bms-formation.com (C.L.); rafael@bms-formation.com (R.Z.-P.); 2Clinical-Based Human Research Department, Foundation Centre for Osteopathic Medicine (COME) Collaboration, 65121 Pescara, Italy; giandomenico@bms-formation.com; 3Research Department, A.T. Still Academy Italia (ATSAI), 70124 Bari, Italy; 4Osteopathy Track and Field Division, Istituto Superiore di Osteopatia, 20126 Milan, Italy; giacomo.consorti@isoi.it; 5Department of Biomedical and Neuromotor Sciences (DIBINEM), University of Bologna, 40126 Bologna, Italy; marco.tramontano@unibo.it; 6Unit of Occupational Medicine, IRCCS Azienda Ospedaliero, Universitaria di Bologna, 40126 Bologna, Italy; 7ESO-Paris Recherche, Ecole Supérieure d’Ostéopathie—Paris, 77420 Champs Sur Marne, France; laurent.stubbe@eso-recherche.fr; 8CIAMS EA 4532, Université Paris-Saclay, 91405 Orsay, France; 9CIAMS EA 4532, Université d’Orléans, 45067 Orléans, France; 10Division of Family Medicine, University of New England College of Osteopathic Medicine, Biddeford, ME 04005, USA; josephine.conte@mainegeneral.org; 11Maine-Dartmouth Family Medicine Residency, Augusta, ME 04330, USA; 12Osteopathic Research Institute, Osteopathie Schule Deutschland, 22083 Hamburg, Germany; tliem@osteopathie-schule.de

**Keywords:** osteopathic medicine, osteopathic manipulative treatment, manual therapy, professional identity, interprofessional relations, medical rationalities, shared decision-making, therapeutic alliance, enactivism, allostasis, touch, synchrony, interoception, autonomic nervous system, traditional medicine, person-centered care, evidence-based practice

## Abstract

Background. A major goal for a significant portion of the osteopathic community is to update osteopathic principles, satisfying three needs: sourcing from the origin, proposing original and unique practical approaches, and describing the entire process in a scientifically updated way. On this line, several interprofessional proposals for healthcare providers have already been made by implementing patient-centered care and touch-based strategies informed by the enactive model. Enactivism principles can provide a foundation for rethinking osteopathic care by integrating environmental, psychological, social, and existential factors to facilitate the patient’s biobehavioral synchronization with the environment and social context, address health needs, and enhance the quality of multiprofessional healthcare services. However, there is a need to develop a conceptual model that offers a framework for organizing and interpreting disciplinary knowledge, guiding clinical observation and practical strategies, and defining both interprofessional collaboration and the unique focus of the profession. This scoping review and integrative hypothesis aim to fulfill the need for a more detailed and comprehensive understanding of the distinctive osteopathic care to biobehavioral synchrony, emphasizing both interprofessional collaboration and the profession’s unique competencies. Methods. The present article was developed in accordance with established guidelines for writing biomedical scoping reviews. Results. A total of 36 papers were considered for thematic and qualitative analyses, which supported the integrative hypothesis. Considering the current tenets for osteopathic rational practice, we propose an integrative hypothesis to focus on a practical framework for osteopathic patient biobehavioral synchronization. Patient–practitioner–environment synchronization could be promoted through a four-step process: (1) a narrative-based sense-making and decision-making process; (2) a touch-based shared sense-making and decision-making process; (3) hands-on, mindfulness-based osteopathic manipulative treatment; (4) patient active participatory osteopathic approaches to enhance person-centered care and rational practice. Conclusions and future directions: The proposed model fosters patient–practitioner synchronization by integrating updated traditional osteopathic narratives and body representations into practice, offering a culturally sensitive approach to promoting health, addressing contemporary health needs, and improving inclusive health services. Future studies are required to assess the transferability and applicability of this framework in modern settings worldwide.

## 1. Introduction

Osteopathic care (OC) is a person-centered healthcare discipline that emphasizes the comprehensive nature of the individual, the interplay between the body’s structure and function, and inherent self-regulatory mechanisms [1]. Two related types of osteopathic practitioners (OPs) are present worldwide: osteopathic physicians, who deliver osteopathic medicine, and osteopaths, who provide OC. Osteopathic physicians are statutorily regulated and can obtain a license to practice medicine in 57 countries [1]. They are required to have a medical degree and post-doctoral training, which includes additional education in osteopathic principles and osteopathic manipulative treatment (OMT). In contrast, osteopaths are statutorily recognized as healthcare professionals and regulated by law in 13 countries. Their qualifications range from diplomas to master’s degrees, with most countries requiring at least a bachelor’s degree for new osteopaths. Osteopathic education and training institutions are found in at least 20 countries, with regulated countries mandating continuing professional development, while voluntary registration in other countries typically involves informal requirements. Globally, approximately 196,861 OPs provide OC across 46 countries, with 117,559 being osteopathic physicians or physicians with additional osteopathic training and 79,302 osteopaths. Of these, 45,093 osteopaths are statutorily regulated and registered, while an estimated 34,207 are registered with voluntary organizations. In 22 countries, OC is either not recognized or not regulated by statute, and registration is voluntary [1]. OC makes a substantial contribution to healthcare across the globe. However, within the community of practice, there is an ongoing debate regarding the update of osteopathic theoretical and clinical practice models, with “tradition-dismissive” authors [2,3,4,5,6] on one side and “tradition-reconceptualization” osteopathic practitioners (OPs) on the other [7,8,9,10,11,12,13,14,15,16,17,18,19,20,21,22] (Table 1).

To move beyond this contentious academic debate, we have focused on updating clinical practice models to help practitioners fully utilize the professional skills introduced by the osteopathic profession. A growing consensus is emerging around the adoption of a phenomenologically enactive approach to sense-making, decision-making, and clinical practice [23,24,25,26,27]. Enactivism is a theoretical framework rooted in cognitive science and phenomenology, which posits that cognition arises through dynamic interactions between the body, mind, and environment [24]. Both OC and enactivism emphasize the integral roles of the body and environment in maintaining health. In this context, the environment encompasses the patient’s physical surroundings, social context, agency, and functional interactions [24]. From an enactivist perspective, therapeutic touch and hands-on techniques employed by physiotherapists, OP, chiropractors, and massage therapists are recognized as effective strategies for fostering interpersonal connections. By using the body as a central medium, these techniques used in a panprofessional approach enhance biobehavioral synchrony—the harmonization of individuals in terms of body–mind behavior, including postural, autonomic/physiological responses, endocrine activity, and brain function during social interactions and daily activities within the physical environment [28,29]. This process strengthens therapeutic alliance and improves clinical outcomes [29,30,31,32,33]. In recent qualitative research, besides (1) fostering meaningful dialog, (2) encouraging active patient participation, and (3) synchronization between practitioners and patients has emerged as a category for describing the therapeutic alliance in musculoskeletal care between OC and physiotherapy able to influence the therapeutic process [34]. This emerging trans-professional framework bridges disciplinary gaps, highlighting shared competencies between healthcare professionals (e.g., physiotherapy and OC) [34], such as strategies to improve patient–practitioner biobehavioral synchrony, alignment of behavior, affective states, and biological rhythms (e.g., circadian rhythms like sleep–wake cycles, respiratory rate, heart rate variability) [34], ultimately regulating patient allostasis [35]. 

Moreover, several authors highlight the inclusion of mindfulness, behavioral strategies, and pain education [36,37,38] implemented by non-mental health professionals to address psychological distress and promote health-related behaviors [39,40,41,42,43]. The contemporary international scenario views an OP as a healthcare professional who employs a variety of manual techniques, combined with educational strategies on lifestyle, such as exercise and diet [1]. However, it would be valuable to consider that individuals with similar needs may prefer different approaches (top-down, mixed-approach top-down/bottom-up (or mind/body) hands-on techniques) to achieve synchronization of physiological functions to manage symptoms, enhance health, and maintain overall well-being [12]. Effective interprofessional collaboration between healthcare professionals requires clear communication, mutual understanding, and respect to ensure cohesive patient care [44]. Acknowledging each profession’s distinctive competencies enhances patient outcomes and promotes an integrated healthcare system [44]. Developing a conceptual model offers a structured framework to organize knowledge, guide clinical observation, and refine practice focus [10]. Proposed OC, integrating strategies from other disciplines applied following the osteopathic principles, aims to synchronize osteopath–patient perceptions to foster shared understanding and decision-making [12]. An interprofessional team of clinicians, researchers, and academics recently examined body representations and consciousness to better inform the current principles of an osteopathic model for patient–practitioner–environment synchronization, integrating person-centered care, evidence-based approaches, and traditional epistemological insights. Additionally, the authors advocate for developing a practical framework [22], including distinctive OMT aimed at synchronizing physiological rhythms [45]. The authors aim to incorporate three core aspects of osteopathic care (OC) to structure the practical framework: (1) the application of the osteopathic palpatory diagnosis process (OPDP) and the identification of somatic dysfunction (SD), (2) a holistic approach grounded in osteopathic principles, and (3) the individualization of OC tailored to the unique needs of each patient. Our hypothesis is that drawing inspiration from osteopathic heritage, reconceptualized through evidence-informed knowledge, can lay a foundation for developing innovative and practical approaches within the osteopathic profession. For these reasons, this scoping review and integrative hypothesis paper propose a comprehensive explanation of the distinctive osteopathic approaches to biobehavioral synchrony, highlighting both interprofessional and distinctive competencies.

## 2. Materials and Methods

The present article was developed in adherence to established guidelines for writing a scoping review [46]. It was registered with protocol number osf.io/73fxu on the Open Science Framework Registry. To offer a practical framework within the osteopathic field, the authors analyzed relevant data on distinctive osteopathic approaches to biobehavioral synchrony, emphasizing both unique competencies and interprofessional collaboration. 

The theoretical framework was crafted by a team of experts (G.D’A., C.L., and F.B.) with over 10,000 h of professional experience in education, scientific research, and clinical osteopathic practice [47]. This framework emerges from a collaborative brainstorming process grounded in clinical observations and the best available evidence.

### 2.1. Research Question

“How does the implementation of an osteopathic practical framework for biobehavioral synchrony, incorporating both interprofessional and osteopathic-specific competencies, enhance patient–environment synchronization, address critical health needs, and improve multiprofessional healthcare services, compared to a panprofessional, tradition-dismissive model? A scoping review”.

### 2.2. Search Strategy

A comprehensive literature search was conducted from July to December 2024 using MEDLINE (PubMed). To select the terms to be used in the literature search, the research team reviewed MeSH terms and their relevant subheadings related to OC and other manual therapies [48]. Eleven MeSH terms were deemed effective for retrieving the manual therapy research literature: manipulation, osteopathic, osteopathic medicine, chiropractic, exercise movement techniques, exercise therapy, manipulation, orthopedics, massage, muscle relaxation, muscle stretching exercises, musculoskeletal manipulations, and traction. MeSH terms have been combined with additional search terms pertinent to the focus of the paper, including the following keywords: professional identity, interprofessional relations, medical rationalities, shared decision-making, therapeutic alliance, enactivism, allostasis, touch, synchrony, interoception, autonomic nervous system, traditional medicine, person-centered care, and evidence-based practice. The implemented MEDLINE research strategy is: (“Manipulation” OR “Osteopathic” OR “Osteopathic Medicine” OR “Chiropractic” OR “Exercise Movement Techniques” OR “Exercise Therapy” OR “manipulation” OR “Orthopedic” OR “Massage” OR “Muscle Relaxation” OR “Muscle Stretching Exercises” OR “Musculoskeletal Manipulations” OR “Traction”) AND (“professional identity” OR “interprofessional relations” OR “medical rationalities” OR “shared decision-making”, OR “therapeutic alliance” OR “enactivism” OR “allostasis” OR “touch” OR “synchrony” OR “interoception” OR “autonomic nervous system” OR “traditional medicine” OR “person-centered care” OR “evidence-based practice”). Search terms were tailored to each database, and relevant subheadings were applied where appropriate. The search was conducted without limitations on study design, population, outcomes, or publication date. The reference lists of the identified articles were also reviewed, and a snowball sampling approach was employed to uncover additional relevant studies.

### 2.3. Eligibility Criteria

To capture the full spectrum of available information on the topic, no formal validity or quality assessments were performed on the selected articles. Papers eligibility has been determined through a two-step selection process (abstract and full-text), conducted independently by the team of experts (G.D’A., C.L., F.B.). Additionally, relevant articles were identified by examining the reference lists of eligible studies. This snowball sampling approach was employed to enhance the comprehensiveness of the review and ensure that important studies that were not captured in the initial search were included. The relevance of each article to the study’s focus was critically evaluated through brainstorming sessions among the same experts. Articles were excluded from the screening process if they described the use of manipulative techniques or psycho-corporeal educational and counseling approaches without the application of osteopathic clinical reasoning models and principles. Specifically, studies were excluded if these techniques were used within the context of manual therapy provided by healthcare professionals who were not OPs. Additionally, articles were excluded if they did not align with the three core aspects of osteopathic care (OC) that the authors selected to structure the practical framework for this review. 

These three key aspects include (1) the use of the osteopathic palpatory diagnosis process (OPDP) and of SD, (2) the holistic approach rooted in osteopathic principles, and (3) the individualization of OC based on the unique needs of each patient. Only articles that addressed these criteria were considered eligible for inclusion. The content of the included articles has been analyzed to identify emergent themes and extract additional references relevant to addressing the research question, specifically in describing a practical framework for osteopathic approaches to biobehavioral synchrony. Where applicable, the studies were included in multiple categories.

## 3. Results

The identification of records is summarized in Figure 1 [49]. The results may suggest that biobehavioral synchrony can be easily investigated and/or elicited by top-down (verbal and visual) stimuli [50,51,52]; secondly, in psychological research, body-oriented attention seems to empower biobehavioral synchronization [52,53,54,55]. Twenty-two findings [23,24,25,34,56,57,58,59,60,61,62,63,64,65,66,67,68,69,70,71,72,73] were analyzed and grouped by pertinence (Table 2), and 14 papers were added through a snowballing process [10,12,29,74,75,76,77,78,79,80,81,82,83,84], drawing from previous authors’ knowledge and references of the included records. Finally, 36 papers were considered for thematic and qualitative analyses, supporting the integrative hypothesis. The primary literature search revealed a mismatch between top-down (psychological studies) and bottom-up (manual therapy studies) approaches in terms of number and pertinence regarding biobehavioral synchronization.

Additionally, nearly one-third of the included articles were identified through a snowballing process, rather than using primary bibliographic search strings. This suggests that OC may have inherently integrated the principle of biobehavioral synchronization, although it had not yet been formally articulated using contemporary keywords. It emerges that patient–practitioner biological and behavioral synchronization could be promoted through a four-step process (Table 3): (1) a narrative-based sense-making and decision-making process; (2) a touch-based shared sense-making and decision-making process; (3) hands-on, mindfulness-based OMT; (4) patient active participatory osteopathic approaches (PAOA).

In summary, the 22 findings substantiate a four-step framework, while the 14 articles from snowballing offered valuable insights into the practical application of each step in clinical practice. The 22 articles have been grouped by themes, namely the four steps of the practical framework, as outlined below:(1)N. 14 articles enabled the identification of the first step: narrative-based sense-making and decision-making process [24,25,34,56,57,58,59,60,61,62,63,64,65,66] seems to be central to OC. Both verbal and nonverbal communication enhances therapeutic alliance and shared understanding [56,61,62]. The biopsychosocial and complexity medicine models, including Cynefin frameworks (CF), support tailored interventions based on patient-reported narratives and embodied experiences [57,65,66]. A shared generative model improves predictability, reducing uncertainty in treatment [25,34]. Practitioner-led becomes shared decision-making to refine reasoning processes and adapt treatment strategies [58,60,64]. Sense-making itself has therapeutic value, transforming illness experiences into a coherent framework [63,64,65]. Narrative-based sense-making fosters an embodied therapeutic relationship, integrating clinical reasoning, patient experience, and shared decision-making, reinforcing OC’s distinctiveness in healthcare.(2)N. 16 articles enabled the identification of the second step: touch-based shared sense-making and decision-making process [24,25,34,56,57,58,59,60,61,62,64,65,66,67,68,69]. Touch, implemented in OPDP, is an interactive process that influences body perception and clinical decision-making [34,56]. In OC, SD serves as a tool for communication and participatory sensemaking in the OPDP, enabling immediate patient feedback for targeted decisions [57,58]. Touch guides an iterative decision-making model, integrating clinical history, functional physical examination, and contextual factors [59] to foster a relationship-centered OC [67]. The integration of mindfulness and OPDP enhances biobehavioral synchrony and mental state alignment [23,25]. Touch strengthens the therapeutic alliance [24] and functions as an interpretative and affective practice to understand the patient [60,61]. Touch modulates interoception and pain, supporting body perception and therapeutic engagement [62,64]. Zegarra-Parodi et al. [65,66] highlight the importance of cultural adaptation and osteopathic narrative. Vismara et al. [68] highlight the importance of utilizing patients’ physiological responses in the OPDP (e.g., child stress behavior) to better understand their tolerance to OMT. Touch as a tool for shared sense-making enhances interaction, clinical reasoning, and personalized care, integrating bodily signals, interoception, and therapeutic synchrony.(3)N. 20 articles enabled the identification of the third step: hands-on, mindfulness-based OMT [23,24,25,32,34,56,57,58,59,60,61,62,63,64,65,66,69,70,71,72,73]. OC blends OMT with mindfulness to enhance interoception, body awareness, and synchrony. Affective touch activates C-tactile fibers, promoting autonomic and neurochemical responses [56,71]. OP adapts touch based on tissue feedback, refining bodily perception [58]. Combining touch with mindfulness improves self-regulation, pain relief, and interoceptive accuracy [23,60,63]. This interactive loop helps interpret bodily signals [25,61] and builds therapeutic connections [34,66]. Integrating OC with mindfulness supports comprehensive well-being, especially for interoceptive overload across patients’ conditions (23,62).(4)N. 14 articles enabled the identification of the fourth step: PAOA [23,25,56,57,58,59,60,61,62,63,64,65,66,70] emphasize patient engagement, body awareness, and self-management. Patients are encouraged to participate in their treatment through movement, mindfulness, and experiential bodywork [58,59,60]. Communication, education, and psychological support enhance patient agency and perception of health [23,25]. Techniques like guided breathing exercises supported by touch, experiential bodywork, and lifestyle adjustments help integrate interoceptive, proprioceptive, and exteroceptive awareness [62,63,64]. This person-centered model combines manual therapy with behavioral and cognitive strategies for adaptive self-care [57,66].

The findings, supplemented by the papers included through the snowballing process [10,12,29,74,75,76,77,78,79,80,81,82,83,84], provide the basis for developing an integrative hypothesis that outlines a practical four-step osteopathic process aimed at enhancing synchronization between the patient and practitioner (for a comprehensive understanding of the eligibility process and reproducibility; see Appendix A: Included articles, relevant content, related emergent themes, and additional references).

## 4. Discussion

This scoping review proposes a practical framework to reconcile conflicting findings in the ongoing debate on updated conceptual and clinical osteopathic models. On one side, there are “tradition-dismissive” authors [2,3,4,5,6], while on the other, “tradition-reconceptualization” osteopathic practitioners (OPs) [7,8,9,10,11,12,13,14,15,16,17,18,19,20,21,22]. To address these conflicting findings, we identified the sources of discrepancy, discussed their implications, and offered possible explanations for the differences. We acknowledged these discrepancies and explored potential points of convergence through an enactivist perspective. While this review may not resolve the conflict, it transparently addresses these variations, transforming the debate into a potential dialog on the evolving perspectives in contemporary osteopathic care. By following integrative hypotheses, we propose a practical framework for osteopathic approaches to biobehavioral synchrony. Finally, we offer suggestions for future research to help resolve contradictions in the existing literature.

*Integrative hypotheses: Patient–practitioner–environment synchronization.* In the following subsections, we present a rational and practical framework for OPs grounded in osteopathic tenets and contemporary insights into body representation and consciousness (Figure 2).

This framework aims to enhance patient biobehavioral synchronization by fostering a sense that emerges from patient–practitioner relational processes distributed across the mind, brain, body, and environment, with the patient’s experience at the core (Figure 3). This approach focuses on aligning body–mind behaviors, including postural, physiological, and psychological regulatory processes, to support daily activities and social interactions within the physical environment.

The discussion delves into the role of interprofessional collaboration among healthcare professionals while highlighting the unique osteopathic competencies in approaches to biobehavioral synchrony that underpin this proposed four-step process. For a clearer understanding of the clinical applications of the proposed model, see Appendix A: patient–practitioner–environment synchronization— A clinical scenario exemplifying the four-step practical framework.

### 4.1. Narrative-Based Sense and Decision-Making Processes

Early OPs used evocative storytelling to explain complex anatomical and physiological concepts, enhancing mental imagery for practitioners and patients alike [85]. A notable example is Dr. W.G. Sutherland’s “A Tour of the Minnow”, an imaginative journey through the brain cavities that guided attention to key anatomical landmarks [86]. This tradition continues in OC, integrating narrative medicine and patient-centered care to understand health experiences and guide treatment decisions [81]. Modern OC embraces a phenomenological approach, emphasizing the patient’s lived experience and lifeworld in the healing process [66]. To navigate complex patient narratives, the CF has been proposed as a tool for managing internal contradictions and unexplained body representations, fostering a shared understanding between the practitioner and patient [12]. The CF integrates psychological and existential factors with biomedical knowledge, enabling collaborative clinical decision-making [66]. The CF is used to visually structure patient narratives and decision-making, helping patients reframe unclear elements of their health journey. The OP uses a whiteboard or sheet of paper to graphically represent the sense-making and decision-making processes facilitated by the CF [66]. Practitioners encourage patients’ self-reflection to observe unclear elements of health processes from different perspectives and improve sense-making. This process enhances their ability to understand prior treatments and actively participate in OC. “Two-Eyed Seeing” communication bridges illness (subjective experience) and disease (biomedical alterations), allowing for a personalized osteopathic approach that integrates symptom-oriented manual therapy with broader health perspectives [66]. OPs employ a unique decision-making process that incorporates pattern recognition through active and passive patient movement and tactile stimulation of SD-related regions [66,81]. 

This neuroaesthetic enactive paradigm (NEP) enables practitioners to interpret nonverbal responses and verbal cues to select individualized treatment strategies [60]. The CF further supports osteopathic decision-making by guiding the application of evidence-based findings (simple–complicated domains) and personalized, hypothesis-driven strategies (complex–chaotic domains) within the patient–practitioner relationship [12,66,81]. Narrative communication within the CF framework enhances interdisciplinary collaboration by structuring patient stories into comprehensible, evidence-informed formats. OPs contributes a distinctive perspective to interprofessional healthcare teams, using visual narratives to improve patient comprehension and retention of health information [87]. Cognitive learning theories support the effectiveness of combining visual and narrative-based approaches with lived body experiences to enhance shared decision-making and foster integrated patient care [87]. In alignment with the enactivist perspective, OPs use touch to engage the patient’s body, promoting intentional exploration, information processing, and enhanced sense-making [88].

### 4.2. Touch-Based Shared Sense and Decision-Making Processes

A.T. Still, the founder of osteopathic medicine, emphasized that healing is a natural process driven by the body itself [89]. He regarded OMT as a means to restore natural movement, improve circulation and fluid dynamics, optimize nerve function, and support the body’s intrinsic self-regulation mechanisms [90]. Building on Still’s legacy, early OPs developed the concept of SD to enhance the interdependence of structure and function, improve body awareness, and achieve better health outcomes [91]. OPs support patients in understanding their health in relation to their level of self-awareness [66]. Osteopathic encounters, often resembling ritual-like experiences, help patients make sense of their bodies. The re-interpretation of a patient’s body and self-image—shaped by unspoken communication and OMT—is essential for therapeutic interaction. Through multisensory, emotional, moral, and esthetic components, these rituals transform illness narratives into lived experiences, allowing for the recognition of emerging patterns [92]. This shared interpretative process enables practitioners and patients to collaboratively determine effective healing strategies, similar to how contemporary OP applies the NEP [60]. In the osteopathic shared decision-making process, touch is a key communication tool that helps tailor treatment by facilitating interaction, validating clinical findings, and identifying areas of overactive body function [64]. Neuromyofascial active regions act as an interface between practitioner and patient, transmitting the biological and physiological effects of touch [58]. In light of the NEP, OPDP is viewed as a dynamic verbal and non-verbal dialog between patient and practitioner, aimed at generating a positive surprise and significant prediction error, thereby challenging existing beliefs and updating the brain’s generative model [60]. Following the OPDP, the OP identifies an area of interest shared by both the patient and them [11,12,58,79]. This process involves generating a hypothesis based on a mutual perception of discomfort upon touch and a lack of integration of the area in relation to surrounding regions. The OP then tests this hypothesis, either confirming or refuting it through a structured function correlation test procedure [11,12,59,60,64,79,80,81]. In other words, they use a combination of touch, active movements, and patient positioning—typical of OMT—to assess changes in functional capacity, including daily movements and clinical examinations. Positive changes help guide the shared decision-making process for a personalized approach, while negative outcomes lead the practitioner to select strategies based on research in similar clinical contexts [60,78,80]. This shared decision-making process relies on proximity and non-verbal methods, particularly touch, and is reinforced by verbal communication with patients or caregivers. 

A pleasant sensation from specific touch in an SD-related area, emerging from a shared neuroaesthetic enactive experience, may enhance non-specific effects like the placebo response, optimizing therapeutic touch. This process helps patients make sense of their disorders, overall health, and performance [60,78,80]. Additionally, it clarifies the osteopathic approach, especially when manipulative techniques involve distant areas from the symptomatic region, preventing misunderstandings that could lead to negative perceptions, beliefs, or expectations [60,78,80]. This model aligns with person-centered care principles for managing musculoskeletal pain, emphasizing meaningful connections through body awareness [60]. It helps patients understand their condition and treatment by enhancing their connection to their body and using touch to bridge gaps in comprehension. Here, the biological aspect of the biopsychosocial model goes beyond diagnosis and treatment—it becomes crucial for fostering connections that patients perceive as essential for a successful clinical experience [60]. Recent qualitative research highlights that the NEP helps interpret patient responses, actively engaging them in shared decision-making, goal setting, and realistic treatment planning [34]. By incorporating both verbal and non-verbal elements through a neuroaesthetic enactive experience, patients gain a deeper understanding of therapeutic touch and mindfulness-based OMT, fostering synchronization between patient, practitioner, and environment. Like chiropractors and physiotherapists, OPs use OMT to meet patient expectations while also serving as a channel for nonverbal communication, transmitting signals that modulate pain and regulate emotions [25]. Although this shared proximity-based strategy is common across healthcare disciplines, NEP provides a unique osteopathic perspective, where OPDP and intervention rooted in SD act as a bottom-up communicative entry point from the patient’s skin to the central nervous system, consciousness, and body representation [58,60]. Through verbal and nonverbal dialog, this process aims to update the patient’s brain generative model from the OPDP stage onward while enhancing their understanding of hands-on mindfulness-based OMT [60].

### 4.3. Hands-On Mindfulness-Based OMT

Pioneering OPs started their healthcare revolution by merging elements of touch-based therapeutic traditional practices, such as bone-setting, with early hypnosis and mindfulness strategies, such as magnetic healing, into a unified doctrine [16,93]. OMT, as well as other musculoskeletal care strategies, involves a phase of patient–practitioner synchronization aimed at improving therapeutic touch (i.e., functional tactile stimulus) by continuous coordination and adaptation through feedback [34]. In cases where the OPs implement interprofessional strategies to achieve optimal therapeutic alliance through biobehavioral synchrony, they take on a distinctive character [66]. In hands-on and mindfulness-based OMT, practitioners incorporate the concept of SD by focusing on areas of the body or generalized patterns that require both minimalist and maximalist therapeutic intervention [91] and concentrate on natural physiological movements, associated metabolic activity, and energetic vitality [94] to promote patient–practitioner synchronization. Hands-on mindfulness-based interventions have been extensively studied in the last two decades, with growing evidence supporting their efficacy in managing chronic pain, recurrent depression, and addiction [95]. Additionally, increasing interest in multimodal mind–body therapies—such as OMT combined with pain neuroscience education and clinical hypnosis—highlights their potential role in addressing chronic pain and disability [73]. Meditative practices, mindfulness [35], and culturally sensitive OC [66] share the goal of promoting interpersonal and inherent biobehavioral synchrony, as well as tuning with the environment and social context [76]. Hands-on, mindfulness-based osteopathic approaches play an essential role in contemporary OC [38,74,96], as demonstrated by the biodynamic osteopathic approach [97]. 

The incorporation of hands-on mindfulness and meditative exercises within osteopathic practice through (interoceptive) touch-based osteopathic approaches promotes patient–practitioner synchronization and facilitates more sustainable treatments for the patient [74]. OPs, applying hands-on and mindfulness-based approaches, consider SD as a “soma–physiology–experience–context–dysfunction pattern” to favor patient–practitioner–environment synchronization [63,74,76]. There are osteopathic approaches that combine manual therapy with mind–body interventions, such as psychosomatic osteopathy, osteopathic heart-focused palpation, and bifocal integration techniques [63,74]. Notably, the bifocal integration method integrates hands-on OMT with rhythmic eye movements to enhance attentional processes and alleviate stress [76]. The practitioner employs a range of interprofessional therapeutic strategies, including meditative exercises, mindfulness techniques, antalgic positioning, and synchronized music, as preparatory activities aimed at improving patient awareness of illness experiences and desired health outcomes. Additionally, OP utilizes distinctive tactile and sensory stimuli, specifically targeting movement patterns associated with SD, identified through shared decision-making and NEP [60]. By integrating touch-based techniques with rhythmic eye movements, OP facilitates awareness of arousal levels, pain perception, and physiological adaptation, helping patients achieve relaxation and synchronization [63,76]. A key component of hands-on mindfulness-based osteopathy is the use of tactile and perceptual processes to optimize coordination and synchronization between different body regions, thereby enhancing mechano-physiological function. General osteopathic treatment exemplifies this principle, employing interoceptive–proprioceptive rhythmic touch-based stimulation to promote global bodily synchronization [98]. The underlying mechanism of this approach aligns with entrainment-based interventions, which synchronize whole-body rhythms to strengthen the mind–body connection, promote coordinated, fluid mobility, and optimize force transmission—ultimately enhancing the mechano-physiological synchrony of the patient’s structure and function [99]. To achieve the patient’s physiological entrainment, all contextual factors—including environmental conditions, verbal communication, and osteopathic touch—play a crucial role in fostering a state of mindfulness in both the patient and the practitioner. OPs must develop their own bodily awareness and cognitive self-regulation to ensure that their touch-based interventions foster patient relaxation and autonomic balance. Additionally, they guide the patient’s attentive focus toward breathing patterns and the anatomical regions perceived as more relaxed during treatment [76]. Anatomy plays a strategic/crucial role in OC, enhancing mental imagery and focused attention in both practitioners and patients. This cognitive framework allows patients to develop a more positive anatomical and physiological perception of their health status [85]. The ability of OP to detect subtle physiological changes—such as tissue tension, blood flow variations, and respiratory adaptations—reinforces the role of touch-based interventions in promoting health [100]. Studies suggest that osteopathic touch-based mindfulness techniques may regulate biological oscillators, including heart rate variability, vasomotion, and breathing patterns, contributing to allostatic balance and therapeutic efficacy [45,101]. OPs promote mindfulness-based hands-on approaches, focusing on tissue physiological rhythm changes to enhance coordination and synchronization effects, balancing the autonomic nervous system [45]. A hermeneutic model for cranial osteopathy suggests that osteopathic touch-based mindfulness fosters a shared therapeutic relationship, helping both practitioner and patient make sense of health processes and illness, thus improving the patient’s experience of well-being [102]. These approaches leverage the affectivity of touch to facilitate the understanding of symptoms and direct interventions toward SDs [61]. OC also involves a bottom-up, touch-based strategy of non-verbal communication, using mechanosensitive tissue interactions to promote synchronization and regulate allostasis. 

Practitioner and patient perceive and modulate changes in allostatic load, such as tissue stiffness, elasticity, and breathing patterns, until normalization occurs [64]. Palpation with focused attention on physiological rhythms (e.g., visceral motion-focused palpation) improves clinical outcomes [63,103]. The OPs’ sustained tactile attention influences functional connectivity in brain regions associated with interoception and attention [71]. Additionally, interoceptive and proprioceptive cues guide coordination and adaptation through patient feedback on body awareness and movement patterns. In some cases, OMT integrates NEP, combining palpation with visual (eye movement), auditory, and rhythmic tactile stimuli aligned with breathing. Patients focus on sensations in palpated areas, sometimes linked to psychosocial stressors, until achieving relaxation marked by regulated breathing, heart rate, and muscle tone [74]. This process enhances interoceptive [72] and proprioceptive integration [104], brain connectivity [105], and clinical outcomes [106]. Once patient–practitioner–environment synchronization is achieved—marked by patient relaxation—the OP applies a personalized tactile stimulus to enhance musculoskeletal movement, patient agency, and regulatory system function [60]. Beyond mindfulness-based osteopathic approaches, a participatory model incorporates goal setting, experiential bodywork, and motivation to improve therapeutic outcomes. These methods share traits with body-oriented psychological therapies [107] and mind–body practices like yoga [108]. This approach is distinct to OC due to its diagnostic link with SD, which guides intervention and reintegration into interoceptive and proprioceptive processes [64]. SD severity correlates with physiological markers, chronic pain, and disability, influencing treatment duration [68,109,110,111]. Osteopathic manipulative procedures, though similar to other manual therapy techniques, remain unique in their integration of neuroaesthetic enactive experiences within patient–practitioner interactions [60]. Through individualized OMT, OPs facilitate whole-body movement, regulatory system function, and patient agency, enhancing clinical outcomes and overall health perception. As OC evolves, the continued exploration of mindfulness-based and touch-centered approaches offers significant potential for advancing patient-centered care within the profession.

### 4.4. Patient Active Participatory Osteopathic Approaches

Early OC aims to foster self-care [15], now referred to as the internal locus of health, also by integrating elements of traditional medicine epistemologies [65,94]. Internal locus of control is an individual’s belief in their ability to control health outcomes by emphasizing personal responsibility and proactive health behaviors [112]. Promoting an internal locus of control is essential for achieving better clinical outcomes [113]. Indeed, since its origins, OC has blended passive manipulative techniques with active patient involvement and lifestyle recommendations, including diet, exercise, and leisure activities [81,114]. Engagement occurs both during the application of methods and through supportive behavioral activities that complement treatment [81,84]. This approach addresses not only structural and physical dynamics but also emotional and psychological states. It encourages active patient involvement in the healing process, employs a phenomenological perspective to explore the “lived body”, integrates and releases dysfunctional emotional and psychological energies, and embraces the concept of stillness to enhance therapeutic synchronization and awareness. The clinical goal involves stress management and self-care to support a comprehensive integration of body, mind, and spirit, incorporating elements from mindfulness and hypnosis techniques applied during manipulative treatments [115]. Early PAOAs often engage patients’ body awareness by using osteopathic touch to encourage them to perceive favorable movement patterns [81,84]. Following this, patients are asked to engage in active motion to experiment with suggested new functional schemas [116], including the use of exercise straps to guide muscle contractions and movements [117]. 

According to a 2020 report on the global status of the osteopathic profession published by the Osteopathic International Alliance in collaboration with the World Health Organization, OPs, true to their origins, continue to employ a variety of manual techniques and frequently provide advice on lifestyle, exercise, activity, diet, and ergonomics [1]. This underscores that OPs, like other health professionals, promote patient health education through self-care strategies that integrate multiprofessional competencies, such as yoga principles [107], synchronized music listening [70], and essential oils [118], guided by specific prerequisites and training [119]. Simultaneously, active and participatory approaches deeply rooted in osteopathic principles help define the profession’s distinctive character. A group of clinicians and researchers recently proposed an assessment method for OPs (and other professionals) and self-assessment for patients to evaluate functional motor abilities with a simplified scoring system while performing a movement body scan (concentration on bodily sensations) [84].

The so-called functional neuromyofascial activity (FNA) applied in OC is not intended to replace the existing motor skill evaluation models commonly shared among healthcare, movement, and sports science professions [120,121,122,123]. These include the assessment of conditional abilities (e.g., strength, speed, endurance, and joint mobility) and coordination capacities (e.g., orientation, balance, agility, sense–movement coordination, and reaction). FNA allows both the agent and the OP (and other professionals) to be aware of the local and global compensatory movement patterns characterized by dysfunctional movements in specific body regions or across the whole body [84]. How the FNA was structured also aims to make somatic aspects associated with alterations in motor function (i.e., SD) perceptible and understandable to patients and healthcare professionals who are not trained in the use of diagnostic–therapeutic touch. Moreover, FNA offers valuable insight into the clinical reasoning behind the application of different types of touch, including in distant areas of the body. The proposed method includes “FNA-snacks” or daily routines: time-efficient and well-tolerated strategies of functional activities performed periodically or daily [84]. In cases where the patient is unable to perceive improvements (even those shown by the scoring system), they can refer to their OP, who can then apply principles of OMT (e.g., integrated neuromusculoskeletal release [116]). Specifically, the OP incorporates OMT with the patient in motion while simultaneously facilitating the FNA procedure. This active neuromyofascial release, incorporating FNA interprofessional experiential body work delivered in an osteopathic manner, utilizes OMT applied to the patient in motion. This promotes a strategy to enhance the patient’s ability to synchronize their movements with their environment. PAOA, such as walking FNA, integrates conscious walking [124,125] with mindfulness-based body scans [126] and OMT (also applied during patient movement) [84] to promote healthy aging, enhance mobility, and encourage patient adherence to physical activity and self-management while incorporating interdisciplinary knowledge on nutrition and exercise to support referrals to other health professionals [127,128] (see Appendix B. Patient active participatory osteopathic approaches across different life stages; Section A.1. Patient active participatory osteopathic approaches across different life stages). OC for children, promoted by OP with specific prerequisites [129], includes PAOA [81,84] that integrate osteopathic principles with developmental psychology [130] and neuropsychomotor rehabilitation.Using OMT [75] focused on SD and whole-body compensatory patterns [131], movement, and family involvement, this approach promotes motor coordination, emotional regulation, and healthy development while fostering synchronization within the parent–child–OP triad [56] (see Appendix B. Patient active participatory osteopathic approaches across different life stages; Section A.2. Patient active participatory osteopathic approaches in the pediatric population).

### 4.5. Interprofessional Collaboration and the Unique Competencies Underpinning the Proposed Four-Step Process for Achieving Patient–Practitioner–Environment Synchronization

In the patient–osteopathic-practitioner–environment synchronization framework, special emphasis is placed on the environment, given its crucial role in healthcare practice in shaping the therapeutic relationship, influencing patient perceptions, and impacting treatment outcomes [132,133,134]. The environment acts as a co-regulator in biobehavioral synchronization, influencing the interaction between the patient and practitioner. Optimizing environmental conditions can enhance treatment effectiveness by aligning sensory, physiological, and psychological processes to support patient-centered care. This is particularly significant in OC, where the patient’s perception of a safe environment—facilitated by NEP within the OC context—enhances their internal locus of control, a key factor for active participation in their care [60]. On the one hand, the environment impacts the biomechanical and ergonomic aspects of posture, gait, and movement coordination. On the other hand, the environment provides cues for motor planning, spatial awareness, and functional integration, which are critical in OC. OMT, both passive and active participatory, serves as a bridge between the practitioner and patient for biobehavioral synchronization. OPs gradually expose the patient to interaction with the environment and contextual factors, first by passive, then active participatory OMT, to better integrate patient emergent sensory, autonomic, emotional, and motor functions. It fosters neuromodulation, therapeutic alliance, and sensorimotor alignment, making it a crucial element in osteopathic care. Regarding the narrative-based sense and decision-making step, OP promotes the application of a framework rooted in complexity science [135]. This approach leverages interprofessional tools and language to navigate the complexities of modern healthcare in collaboration with other professionals. Traditional decision-making models rooted in Industrial Age principles often fall short in addressing these complexities. As a solution, healthcare professionals, including nurses, have proposed the use of the Cynefin sense-making framework as a complex adaptive system to improve decision-making in healthcare. In OC, the CF is applied in a distinct manner, incorporating nonverbal bodily narratives through touch to facilitate the pattern recognition processes characteristic of complex scenarios [66]. To the best of our knowledge, this approach is unique to healthcare professionals. Unlike other healthcare providers, OP establishes a neuroaesthetic enactive context for a patient’s experience [66]. By engaging the patient’s active and passive movements and applying tactile stimulation to body regions associated with SD, they interpret emerging patterns to guide a shared clinical decision-making process [60]. The proposed step for touch-based shared sense and decision-making underscores how OPs such as physiotherapists employ touch strategies to meet patient expectations and communicate nonverbally with the patient’s brain [25]. However, when comparing clinical models of physiotherapy and OC, a distinction can be made in their approach to touch as a decision-making driver. Physiotherapy tends to follow a more standardized model, whereas OC relies on patient-emergent patterns to validate palpatory-based hypotheses. For example, the regional interdependence model of physiotherapy posits that seemingly unrelated impairments in distant anatomical regions may contribute to a patient’s primary symptoms [136]. Clinical practice suggests that interventions targeting one body region can influence remote, seemingly unrelated areas. This concept is applied to musculoskeletal connections, such as those between the foot and ankle and the lumbosacral region in the lower quarter, or the cervical and thoracic spine and shoulder symptoms in the upper quarter [136]. In such analytical clinical reasoning scenarios, decision-making is often guided by a predefined anatomical map. Conversely, when faced with complex and chaotic clinical scenarios, OP shifts to a hypothetical–deductive reasoning approach [66]. They use emergent patient patterns to uncover nonlinear connections between symptoms and the region of interest [58]. 

This approach enriches the nonverbal dialog between patients and OP, fostering a positive surprise that generates a prediction error, challenges existing assumptions, and updates the brain’s generative model [60]. Consequently, while applying hands-on, mindfulness-based, and active participatory approaches, OP, unlike other manual therapists, delivers personalized OMT defined in a neuroaesthetic enactive manner [60]. In addition, OP promotes OMT based on outcomes derived from the best available evidence [59,60]. This symptom-oriented OC is shared among all healthcare professionals, making it part of a comprehensive treatment plan [59]. As such, it is similar to other manual therapies in both name and procedural execution. Conversely, personalized OC remains distinctive, as it is informed by the concept of SD, shaped by NEP [60], and supported by a culturally sensitive epistemological perspective [66]. This patient–osteopathic-practitioner–environment synchronization framework highlights the distinctive nature of osteopath–patient interactions, mediated through the somatic body (i.e., the musculoskeletal system), which serves as an interface for synchronization, allowing for a comprehensive understanding and treatment of whole-body dynamics (i.e., the interplay of host and disease factors in an illness state) [137]. It aims to enhance the integrated function of key physiological stress-regulation systems—i.e., neurological, biomechanical, respiratory-circulatory, metabolic-energetic, and behavioral—by utilizing a combination of bottom-up and top-down strategies [137]. In light of patient–osteopathic-practitioner–environment synchronization, OMT—including more debated approaches such as osteopathy in the cranial and visceral fields—can be reconceptualized as body-centered, mindfulness-based strategies targeting the musculoskeletal system to facilitate neural–visceral–sensorimotor coregulation [58,83]. Ultimately, this approach seeks to prevent illness (primary prevention), detect and mitigate complications (secondary and tertiary prevention), and reduce overmedication (quaternary prevention) [138,139].

### 4.6. Integrating Contemporary Knowledge and Models: Positioning Osteopathic Care Within the Broader Context of Manual Therapy and Integrative Medicine

To the best of our knowledge, within the osteopathic community, there has been a misinterpretation of osteopathic principles and their application in practice [10]. We do not believe that osteopathy has ever been reduced to a “touch-only intervention that disregards the diverse therapeutic approaches supported by growing evidence” as some authors have claimed [3]. Instead, there is a need to develop and disseminate a comprehensive strategy that builds on the points of convergence among the various perspectives emerging in the contemporary academic debate [140]. We acknowledge enactivism as a promising conceptual model for reconciling the polarized approaches that define contemporary osteopathic academia, research, and practice. OC and enactivism are distinct fields; however, they share foundational concepts, particularly in their understanding of the dynamic interplay between the body, mind, and environment [141]. Moving forward, it is crucial to critically reflect on the enactivist-informed perspectives that have been integrated into osteopathic care thus far [141]. Instead of adhering to rigid dichotomies—such as biomedical versus biopsychosocial, hands-on versus hands-off, or practitioner-centered versus patient-centered—osteopathic practice should acknowledge the nuanced and complex realities of clinical care [14,142]. Furthermore, maintaining practical relevance in detecting somatic manifestations remains essential to ensuring that theoretical advancements align with the real-world context of osteopathic practice [143]. 

Therefore, how does OC position itself in the contemporary era concerning the conceptual and practical models that should guide practitioners in their daily practice?

According to the authors of this scoping review, it is possible to develop a vision of contemporary OC that integrates evidence-informed and patient-centered approaches, preserving the distinctive identity of the profession rather than replacing osteopathic principles entirely [3,4,5,6] with approaches that fall strictly within the domain of other healthcare disciplines—such as cognitive–behavioral therapy strategies [36,37,38]. While we acknowledge that such strategies are part of interprofessional competencies, as has long been the case in physiotherapy [39], we argue that contemporary osteopathic care (OC) can preserve its unique character while also integrating contemporary knowledge [65] regarding body representation and the influence of consciousness on health [22]. This approach aims to support individuals and communities through health promotion and management, employing both manipulative and educational strategies to assist the musculoskeletal system and related physiological functions. Firstly, OC can contextualize the proposed osteopathic principles [144] in contemporary practice, as, when correctly applied, they continue to offer valuable insights for a culturally sensitive approach that informs evidence-based practice [65]. The first principle, “a person is the product of dynamic interaction between body, mind, and spirit” [144], allows OP to consider bodily, mental, and existential dimensions as crucial to their adaptive capacity in relation to their environment and social context (i.e., also by implementing allostatic indexes [64]). The second principle, “an inherent property of this dynamic interaction is the capacity of the individual for the maintenance of health and recovery from disease” [144], drives the OPDP in a shared understanding of the interdependence between somatic elements and bodily functions (i.e., SD) through the application of the NEP [60]. The third principle—“many forces, both intrinsic and extrinsic to the person, can challenge this inherent capacity and contribute to the onset of illness” [143]—forms the foundation of the shared functional physical examination, which assesses the individual’s self-regulatory functional capacity [11]. The fourth principle, “the musculoskeletal system significantly influences the individual’s ability to restore this inherent capacity and therefore resist disease processes” [144], underpins the stage of mutual understanding between the patient and OP regarding how OMT and PAOA can stimulate the somatic body and influence health processes [81]. Secondly, OC can contextualize the principles of evidence-informed practice by integrating both the practitioner’s and the patient’s experiences within a personalized OC while also incorporating the best available evidence in relation to the patient’s clinical context to ensure a symptom-centered practice [59]. Additionally, OC aligns with the principles of person-centered care by considering biological, psychological, and social factors in assessing an individual’s adaptive capacity [11]. It further emphasizes shared decision-making processes that utilize bodily representations to enhance the therapeutic alliance, as seen in the NEP [60], by implementing narrative-based and touch-based sense and decision-making processes in the OPDP. By using “the body as a pivot” [145], the neuroesthetic-embodied paradigm (NEP) [60] aligns with person-centered care for musculoskeletal pain and supports the evaluation of the interdependence between somatic elements and bodily functions (i.e., somatic dysfunction), particularly through the implementation of structure–function correlation testing (SFCT). It also emphasizes the assessment of self-regulatory potential as a key determinant of functional capacity. This approach promotes integrated strategies that actively engage the patient through hands-on and mindfulness-based OMT and PAOA [81]. Moreover, contemporary OC maintains its distinct identity by reconceptualizing osteopathic principles in light of new evidence, offering a culturally sensitive practice that views the individual as a dynamic entity encompassing bodily, mental, and existential dimensions, all of which contribute to their adaptive capacity within a given environment and social context [66]. Ultimately, osteopathic treatments informed by these principles facilitate the development of personalized, salutogenic approaches that promote health and well-being.

### 4.7. Limitations and Future Directions

We acknowledge that one limitation of this scoping review is the reliance on a single database, PubMed (MEDLINE), for the literature search. While PubMed is a highly relevant and comprehensive resource for biomedical and healthcare-related literature, including osteopathy and related fields, the use of only one database may limit the breadth of the review. Although PubMed’s robust indexing system and MeSH (Medical Subject Headings) terms facilitated an efficient and targeted search, it is possible that relevant studies from other databases or sources were not captured. To mitigate this limitation, the reference lists of identified articles were manually reviewed, and a snowball sampling approach was employed to uncover additional studies. Future reviews or commentaries could benefit from incorporating multiple databases to enhance the comprehensiveness of the search and capture a broader range of relevant literature. As a second limitation, this paper explores a relatively new research topic with limited prior investigation, which has led to a higher degree of self-citation. Given the emerging nature of this field, much of the existing literature directly addressing the distinctive roles and contributions of osteopathic care (OC) remains sparse. Consequently, the review includes studies authored by the same research team or closely associated with their work. While self-citation is common in nascent fields, it may impact the perceived novelty of the findings. This is compounded by the scarcity of recent publications that specifically focus on the distinctive roles and contributions of OC. In recent years, much of the discourse surrounding OC has centered on its perceived limitations [2,3,4,5,6], particularly in comparison to other healthcare professions, rather than emphasizing its unique strengths and opportunities for development. While significant attention has been given to what OC shares with other professions, less focus has been placed on what the osteopathic community must cultivate to establish a more robust and distinct practice—one that not only integrates interprofessional and transprofessional approaches, essential to all healthcare professions, but also reinforces its unique contributions to addressing health challenges and enhancing healthcare services. The promotion of the four-step framework for patient–osteopathic-practitioner–environment synchronization by OP, integrated with rehabilitation strategies advocated by physiotherapy, can strengthen the therapeutic alliance. This approach enhances patients’ understanding of diagnostics and treatment while emphasizing patient–OP interactions within a person-centered framework [142]. As the body of literature on OC grows, future studies in this area will benefit from a broader range of external sources and perspectives. The authors recognize the need for further discourse to explore and define the distinctive role of OC, solidifying its place within contemporary healthcare systems. There is a need to develop a biobehavioral synchronization model centered on manual therapy—particularly OC—that recognizes the body as a unique vessel for human experience and interconnectedness. This model should emphasize the body’s essential role in adapting to physical, psychological, emotional, and existential stressors, ensuring optimal function and resilience in relation to both others and the environment. By positioning bottom-up, non-verbal, proximity-based mediators—such as touch-based techniques personalized according to SD—as central to osteopathic practice and complementing them with top-down (i.e., verbal) cues, OC’s distinctiveness within a multidisciplinary framework can be more clearly articulated. This approach could play a pivotal role in facilitating patient biobehavioral synchrony and enriching embodied experiences within their environment. Future studies are necessary to assess the transferability and applicability of the proposed framework in contemporary healthcare settings worldwide (Table 4).

## 5. Conclusions

The present scoping review and integrative hypothesis propose patient–osteopathic-practitioner–environment synchronization framework refers to a dynamic, multidimensional process facilitated by OC, in which the patient achieves harmonization across three key interfaces: (1) synchronization with the OP, which serves as a mediator for enhancing awareness of the various multidimensional factors influencing health; (2) synchronization with one’s own body, fostering awareness of psychophysical subjectivity and its impact on well-being; and (3) synchronization with the social and natural environment, promoting awareness of existential and ecological subjectivity in relation to health. It includes interdisciplinary and interprofessional aspects to foster patient–practitioner synchronization (e.g., patient involvement, touch, proximity, nonverbal approach, and effective communication,) integrating renewed traditional osteopathic narratives and body representations into clinical practice. This integrative framework positions the OP as a facilitator of biobehavioral alignment, leveraging osteopathic (ecological) care to enhance self-regulation, adaptability, and overall health resilience. It offers a culturally sensitive and distinctive approach to promoting and maintaining health, addressing contemporary health needs, and improving the quality of inclusive healthcare services. To achieve the expected impact on priority health issues through the proposed framework, the authors emphasize the necessity of collaboration with other healthcare professionals and the support of well-structured and high-standard care systems.

## Figures and Tables

**Figure 1 healthcare-13-00820-f001:**
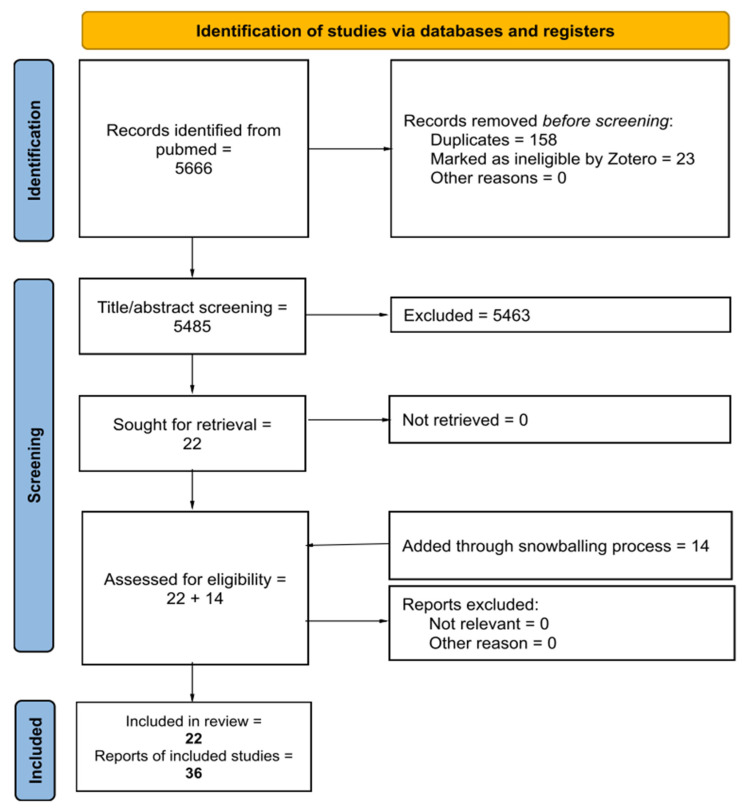
Flow diagram for the selection of studies.

**Figure 2 healthcare-13-00820-f002:**
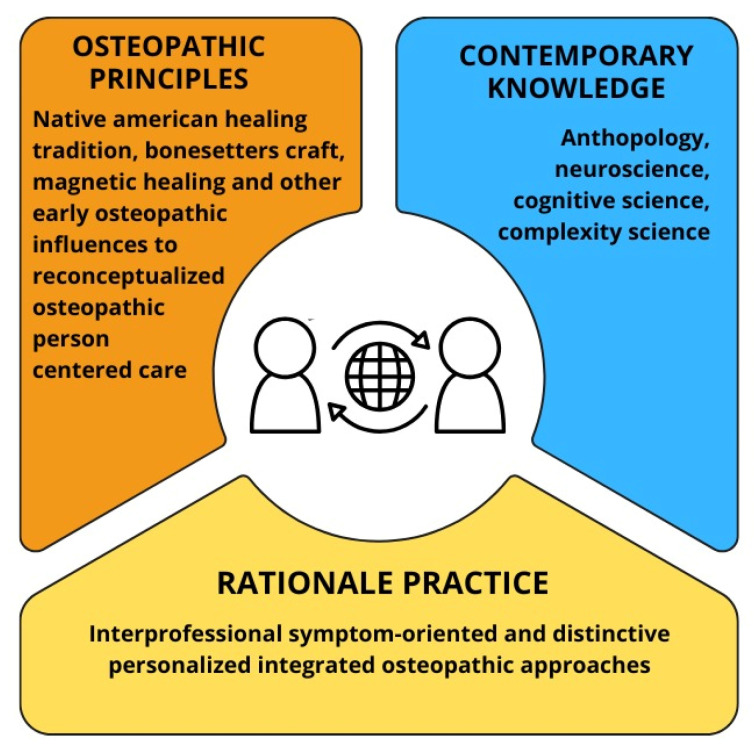
Patient–osteopathic-practitioner–environment synchronization: sources and principles to influence practitioners’ mind-line for contemporary OC.

**Figure 3 healthcare-13-00820-f003:**
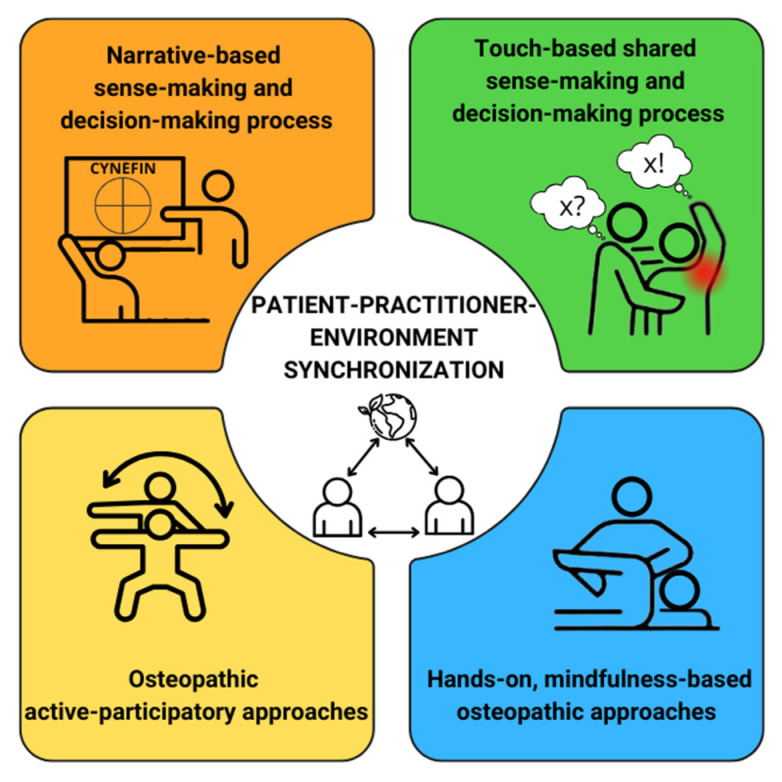
Patient–osteopathic-practitioner–environment synchronization.

**Table 1 healthcare-13-00820-t001:** Debate or dialog: evolving perspectives in contemporary osteopathic care.

Aspect	Tradition-Dismissive Authors	Tradition-Reconceptualization OP
Core Argument	Osteopathic concepts are outdated and mechanistic [2,3].	OC should integrate contemporary science while respecting its foundations [7,8,9,10,11,12,13,14,15].
Criticism of OC	Perpetuates ableist biases [3], nocebo effects, and medicalization of non-specific symptoms [4,5,6].	Supports social equity and maintains osteopathic principles within a modern framework [16,17,18,19,20,21].
View on Tradition	Tradition should be abandoned to align with current scientific paradigms [2,3,4,5].	Tradition should be adapted and updated to remain relevant [7,8,9,10,11,12,13,14,15,16,17,18,19,20,21,22].
Epistemological Approach	Skeptical of historical principles; favors evidence-based practice without traditional influences [2,3,4,5].	Emphasizes epistemological flexibility and the body’s dynamic equilibrium [22].
Theoretical Perspective	Rejects outdated models in favor of fully biomedical or biopsychosocial approaches [2,3,4,5,6].	Advocates for an inclusive approach integrating biological, psychosocial, and existential perspectives [20,21,22].

Abbreviations: OC, osteopathic care; OP, osteopathic practitioner.

**Table 2 healthcare-13-00820-t002:** Included articles, emergent themes, and additional references ^1^.

Included Articles	Theme 1	Theme 2	Theme 3	Theme 4	Relevant References
Accardi et al., 2023 [56]	+	+	+	+	Liem and Neuhuber 2020 [74]; Lunghi et al., 2021 [75]
Arcuri et al., 2022 [57]	+	+	+	+	Liem and Lunghi 2023 [76]; Lunghi et al., 2021 [75]; Lunghi and Liem 2020 [10];
Arrigoni et al., 2024 [34]	+	+	+	-	McParlin et al., 2022a [29]; McParlin et al., 2022b [77];
Baroni et al., 2021a [58]	+	+	+	+	Lunghi and Baroni, 2019 [78]; Lunghi et al., 2016 [79];
Baroni et al., 2021b [59]	+	+	+	+	Sciomachen et al., 2018 [80]; Zegarra-Parodi et al., 2021 [12];
Bergna et al., 2022 [67]	-	+	-	-	
Bohlen et al., 2021 [23]	-	-	+	+	Lunghi et al., 2016 [79];
Cerritelli and Esteves, 2022 [24]	+	+	+	-	
Cerritelli et al., 2017 [71]	-	-	+	-	
Cerritelli et al., 2020 [72]	-	-	+	-	
Consorti et al., 2023 [60]	+	+	+	+	Liem and Lunghi, 2023 [76]; Lunghi et al., 2016 [79]; Lunghi et al., 2022 [81]; Sciomachen et al., 2018 [80];
Elkiss and Jerome 2012 [69]	-	+	+	-	
Esteves et al., 2022 [25]	+	+	+	+	
Groenevelt and Slatman 2024 [61]	+	+	+	+	
Kim et al., 2022 [62]	+	+	+	+	
Liem et al., 2024 [63]	+	-	+	+	Liem and Neuhuber 2020 [74];
Luchesi et al., 2022 [73]	-	-	+	-	
Lunghi et al., 2020 [64]	+	+	+	+	D’Alessandro et al., 2016 [82]; Sciomachen et al., 2018 [80];
Mercadié et al., 2017 [70]	-	-	+	+	
Vismara et al., 2022 [68]	-	+	-	-	
Zegarra-Parodi et al., 2023 [65]	+	+	+	+	Liem and Lunghi, 2023 [76]; Lunghi and Baroni, 2019 [78]; Lunghi et al., 2022 [81]; McParlin et al., 2022a [29]; McParlin et al., 2022b [77];
Zegarra-Parodi et al., 2024 [66]	+	+	+	+	Barsotti, et al., 2023 [83]; Baroni et al., 2023 [84]; D’Alessandro et al., 2016 [82]; Zegarra-Parodi et al., 2021 [12];

^1^ The table reports the articles included in the primary literature search and organizes the relevant contents in 4 themes to inform a 4-step process for patient–practitioner biological and behavioral synchronization: (Theme 1) a narrative-based sense-making and decision-making process; (Theme 2) a touch-based shared sense-making and decision-making process; (Theme 3) hands-on mindfulness-based OMT; (Theme 4) PAOA. The table includes the additional relevant studies emergent from reference lists and the snowball sampling approach.

**Table 3 healthcare-13-00820-t003:** The fourth step of the four-step framework for patient–osteopathic-practitioner–environment synchronization.

Four-Step	Included Articles (n.)	Description
1. Narrative-based sensing and decision-making processes	N. 14 articles enabled the identification of the first step [24,25,34,56,57,58,59,60,61,62,63,64,65,66]	The CF acts as a visual tool to navigate complex patient narratives, categorizing issues into four domains: simple, complicated, complex, and chaotic. OPs employ pattern recognition, expert judgment, and evidence-based approaches to facilitate shared sense-making and inform personalized decision-making. By integrating patients’ lived experiences of illness with the biomedical aspects of disease, they foster a holistic, individualized care model.
2. Touch- based shared sense and decision-making processes	N. 16 articles enabled the identification of the second step [24,25,34,56,57,58,59,60,61,62,64,65,66,67,68,69].	By using the NEP, OPs and patients work together to assess and choose healing strategies through touch. SD-related areas act as a point of interaction for physiological and biological OMT effects. The intensity and type of touch are adjusted based on the patient’s response, enhancing understanding and making the diagnosis and treatment more meaningful. The assessment process involves both verbal and non-verbal communication, with the goal of creating a “positive surprise” that updates the brain’s generative model.
3. Hands-on mindfulness-based OMT	N. 20 articles enabled the identification of the third step [23,24,25,32,34,56,57,58,59,60,61,62,63,64,65,66,69,70,71,72,73].	OMT, along with body–mind synchronization and rhythmic movement techniques, engages both cognitive and sensory-motor processes to promote therapeutic touch through the integration of top-down and bottom-up mechanisms.
4. PAOA	N. 14 articles enabled the identification of the fourth step [23,25,56,57,58,59,60,61,62,63,64,65,66,70]	In the context of PAOA, FNA in osteopathic practice allows OPs and patients to assess motor abilities and identify dysfunctions through a simple scoring system and body scan focused on bodily sensations. This process enhances functional awareness and guides therapeutic decisions. Furthermore, “FNA-snacks” are brief daily routines aimed at improving movement and self-organization, helping individuals track progress and develop strategies for better movement. OPs apply OMT to patients of all ages while in motion (e.g., performing assisted FNA-snacks), facilitating synchronization with the environment.

Abbreviations: CF, Cynefin framework; OC, osteopathic care; OP, osteopathic practitioner; NEP, neuroaesthetic enactive paradigm; SD, somatic dysfunction; OMT, osteopathic manipulative treatment; PAOA, patient active participatory osteopathic approach; FNA, functional neuromyofascial activity.

**Table 4 healthcare-13-00820-t004:** Research roadmap to develop a practical framework for patient–practitioner–environment synchronization.

Research Aim	Research Methodology
To make explicit existing knowledge by observations of “real-world” clinical practice. To provide accurate and transparent data collection from episodes of care, develop testable hypotheses from clinical settings, and inform the delivery of high-quality individualized OC.	Case reports follow CARE guidelines to reduce the risk of bias, increase transparency, and provide early signals of what works for which patients and under which circumstances.
To explore potential clinical values of the proposed practical framework for primary, secondary, tertiary, and quaternary prevention. To evaluate the proposed intervention as a protective factor, plan the interventions, and assess their effectiveness.	Epidemiological studies: case reports, ecological studies, cross-sectional studies, case–control studies, cohort studies, and experimental studies.
To investigate the physiological mechanisms of biobehavioral synchronization.	Real and lab settings’ observational studies.
To implement/preserve the originality of the osteopathic patient biobehavioral synchronization framework among the other manual therapies’ frameworks.	Delphi panel and consensus conference.
To teach the osteopathic patient the biobehavioral synchronization framework.	Mentorship, consensus workshops, continuing professional development.

## Data Availability

Not applicable.

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
