# Peer review of "Patient–Practitioner–Environment Synchronization: Four-Step Process for Integrating Interprofessional and Distinctive Competencies in Osteopathic Practice—A Scoping Review with Integrative Hypothesis"

_healthcare, 2025, doi:10.3390/healthcare13070820_

Round 1

Reviewer 1 Report

Comments and Suggestions for Authors

Dear authors,

Congratulations on your research. It's especially interesting to analyze and update the principles of osteopathic treatment. The following comments are recommendations that could help improve your article.

At the beginning the introduction section, it might be appropriate to point out the legal regulation of osteopathy, which official studies qualify for it, and which healthcare professionals can practice these techniques.

The content of Table 1 is adequate and explanatory, however, it might be advisable to include a table footnote with the abbreviations, although there are few abbreviations used.

The proposed hypothesis is consistent and adequate with the information presented in the introduction, having relevance for clinical practice

Regarding the methodology section, the research question could have been posed in PICOS question format. 

Likewise, although the description of the search terms is adequate, the use of a single database may condition the results obtained and limit the impact of the research.

In the same direction, it could be interesting to clearly define the inclusion and exclusion criteria for screening articles.

It would be advisable to review the information in the flowchart and adjust the format.

The discussion and conclusions appear to be appropriate based on the analyzed topics described in the results section.

Author Response

Thank you very much for taking the time to review this manuscript. Please find below the detailed responses to your comments, along with the corresponding revisions and corrections, which are highlighted in red in the re-submitted files. The revised manuscript now has a word count of 8,000 words, three figures, and four tables. It also includes Appendix 1 and 2, with a total of 1,432 words. The Supplementary Materials consist of "Table S1" (4,583 words), which outlines the eligibility process by detailing the included articles, relevant content, related emergent themes, and additional references. Furthermore, a 5-minute video titled "Video S1: Patient-practitioner-environment synchronization: A four-step practical framework" has been added to enhance readers' understanding of the clinical applications of the proposed model. Additionally, a graphical abstract is provided. Below is a point-by-point answer to the reviewers' comments.

Reviewer 1. Point-by-point response to Comments and Suggestions for Authors

Comments 1: Congratulations on your research. It's especially interesting to analyze and update the principles of osteopathic treatment. The following comments are recommendations that could help improve your article.

Response 1: Thank you very much for your kind words and constructive feedback. We truly appreciate your positive comments on our research, particularly regarding the analysis and update of the principles of osteopathic treatment. We value your recommendations and will carefully address them to further enhance the quality of the article. Once again, thank you for your thoughtful insights. We look forward to submitting the revised manuscript.

Comments 2: At the beginning the introduction section, it might be appropriate to point out the legal regulation of osteopathy, which official studies qualify for it, and which healthcare professionals can practice these techniques.

Response 2: Thank you for your valuable suggestion. In response, we have added the following paragraph (lines 69-86) to the introduction to address the legal regulation of osteopathy, the qualifications required, and which healthcare professionals can practice these techniques: "Two related types of osteopathic practitioners (OPs) are present worldwide: osteopathic physicians, who deliver osteopathic medicine, and osteopaths, who provide osteopathic care (OC). Osteopathic physicians are statutorily regulated and can obtain a license to practice medicine in 57 countries [1]. They are required to have a medical degree and post-doctoral training, which includes additional education in osteopathic principles and osteopathic manipulative treatment (OMT). In contrast, osteopaths are statutorily recognized as healthcare professionals and regulated by law in 13 countries. Their qualifications range from diplomas to Master's degrees, with most countries requiring at least a Bachelor's degree for new osteopaths. Osteopathic education and training institutions are found in at least 20 countries, with regulated countries mandating continuing professional development, while voluntary registration in other countries typically involves informal requirements. Globally, approximately 196,861 OPs provide OC across 46 countries, with 117,559 being osteopathic physicians or physicians with additional osteopathic training, and 79,302 osteopaths. Of these, 45,093 osteopaths are statutorily regulated and registered, while an estimated 34,207 are registered with voluntary organizations. In 22 countries, OC is either not recognized or not regulated by statute, and registration is voluntary [1]. OC makes a substantial contribution to healthcare across the globe."

We hope this addition clarifies the points you raised and enhances the overall clarity of the introduction.

Comments 3: The content of Table 1 is adequate and explanatory, however, it might be advisable to include a table footnote with the abbreviations, although there are few abbreviations used.

Response 3: Thank you for your helpful suggestion. In response, we have added the following footnote to Table 1 to clarify the abbreviations used: "*Osteopathic care (OC); #Osteopathic practitioner (OP)." We hope this improves the clarity of the table for readers.

Comments 4: The proposed hypothesis is consistent and adequate with the information presented in the introduction, having relevance for clinical practice

Response 4: Thank you for your positive feedback. We are glad to hear that the proposed hypothesis is considered consistent and adequate with the information presented in the introduction, and that it holds relevance for clinical practice. We appreciate your support in this regard.

Comments 5: Regarding the methodology section, the research question could have been posed in PICOS question format. 

Response 5:

Thank you for your insightful comment. In response to your suggestion, we have revised the research question using the PICOS format as follows:

P (Population): Osteopathic practitioners and their patients
I (Intervention): Implementation of distinctive osteopathic approaches to bio-behavioral synchrony and integration of interprofessional and osteopathic-specific competencies
C (Comparison): A panprofessional, tradition-dismissive model
O (Outcomes): Patient-environment synchronization, addressing critical health needs, and improving multiprofessional healthcare services
 S (Study Design): Scoping review

The full version of the research question (lines 167-171)is:
"How does the implementation of an osteopathic practical framework for bio-behavioral synchrony, incorporating both interprofessional and osteopathic-specific competencies, enhance patient-environment synchronization, address critical health needs, and improve multiprofessional healthcare services, compared to a panprofessional, tradition-dismissive model? A scoping review of existing literature.."

This format aligns the research question with the scoping review methodology and provides a clearer framework for examining the topic. We believe this approach enhances the methodological rigor and offers a structured way to address the research objectives. Thank you again for your suggestion.

Comments 6: Likewise, although the description of the search terms is adequate, the use of a single database may condition the results obtained and limit the impact of the research.

Response 6: Thank you for your comment. We acknowledge the limitation regarding the use of a single database in our search strategy. As noted in the Limitations and Future Directions section (4.1.6), we recognize that relying solely on PubMed (MEDLINE) for the literature search may limit the breadth of the review. While PubMed is a highly relevant and comprehensive resource for biomedical and healthcare-related literature, including osteopathy, the use of only one database may result in the exclusion of relevant studies from other databases or sources. To mitigate this limitation, we manually reviewed the reference lists of identified articles and employed a snowball sampling approach to uncover additional relevant studies. However, we agree that incorporating multiple databases in future reviews could further enhance the comprehensiveness of the search and capture a broader range of relevant literature. We added the following paragraph in lines 690-701: “We acknowledge that one limitation of this scoping review is the reliance on a single database, PubMed (MEDLINE), for the literature search. While PubMed is a highly relevant and comprehensive resource for biomedical and healthcare-related literature, including osteopathy and related fields, the use of only one database may limit the breadth of the review. Although PubMed's robust indexing system and MeSH (Medical Subject Headings) terms facilitated an efficient and targeted search, it is possible that relevant studies from other databases or sources were not captured. To mitigate this limitation, the reference lists of identified articles were manually reviewed, and a snowball sampling approach was employed to uncover additional studies. Future reviews or commentaries could benefit from incorporating multiple databases to enhance the comprehensiveness of the search and capture a broader range of relevant literature.

Comments 7:  In the same direction, it could be interesting to clearly define the inclusion and exclusion criteria for screening articles.

Response 7: We sincerely appreciate the reviewer's insightful comment. In response, we have now clearly defined the inclusion and exclusion criteria for screening articles by incorporating the following paragraph into Section 2.3, Eligibility Criteria:"Articles were excluded from the screening process if they described the use of manipulative techniques or psycho-corporeal educational and counseling approaches without the application of osteopathic clinical reasoning models and principles. Specifically, studies were excluded if these techniques were used within the context of manual therapy provided by healthcare professionals who were not OPs. Additionally, articles were excluded if they did not align with the three core aspects of osteopathic care (OC) that the authors selected to structure the practical framework for this review. These three key aspects include: (1) the use of the osteopathic palpatory diagnosis process (OPDP) and somatic dysfunction (SD), (2) the holistic approach rooted in osteopathic principles, and (3) the individualization of OC based on the unique needs of each patient. Only articles that addressed these criteria were considered eligible for inclusion."

We believe this addition enhances the transparency and rigor of our methodology. Thank you again for your valuable suggestion.

Comments 8:  It would be advisable to review the information in the flowchart and adjust the format.

Response 8: Thank you for your valuable suggestion. We have carefully reviewed the information in the flowchart and have adjusted the format to improve clarity and readability. We appreciate your feedback, which has helped us enhance the presentation of our study selection process.

Comments 9:  The discussion and conclusions appear to be appropriate based on the analyzed topics described in the results section.

Response 9: We sincerely appreciate the reviewer’s positive feedback. We are glad to hear that the discussion and conclusions are considered appropriate based on the analyzed topics in the results section. Thank you for your insightful evaluation and for recognizing the coherence of our findings.

Reviewer 2 Report

Comments and Suggestions for Authors

This manuscript explores the concept of patient-practitioner-environment synchronization in osteopathic care through a four-step process that integrates interprofessional collaboration and distinctive osteopathic competencies. The study aims to provide a theoretical and practical framework for osteopathic practice, informed by a scoping review of existing literature. The authors propose a model that combines narrative-based decision-making, touch-based shared sense-making, mindfulness-based osteopathic manipulative treatment (OMT), and patient-active participation to enhance bio-behavioral synchrony.

Abstract: Clearly articulates the purpose and scope of the study. Provides a well-structured summary of the proposed framework.

Introduction:

Provides a strong rationale for reconceptualizing osteopathic principles. Engages with contemporary debates on tradition versus modernization in osteopathy. Positions the study within the broader context of interprofessional healthcare.

However, some sections are overly detailed, making the introduction lengthy. The transition to the proposed four-step framework is not well signposted.

Materials and Methods:

Follows established guidelines for scoping reviews. Provides a transparent search strategy, including database use and keywords. Describes the process of thematic and qualitative analysis.

However, the selection criteria for articles are not clearly justified. The rationale for including snowball-sampled studies is not explicitly stated. No discussion of potential bias in study selection.

Results:

Presents findings in a well-organized manner. Groups results by the four-step framework, making interpretation easier. Provides tables and figures to summarize key themes.

However, no mention of how conflicting findings were handled. Tables could include more quantitative indicators to support thematic analysis.

Discussion:

Effectively contextualizes findings within osteopathic and interprofessional healthcare literature. Proposes a novel framework that integrates tradition with modern practice. Highlights potential clinical applications.

However, does not sufficiently address the limitations of the review. Lacks direct comparisons with similar models in manual therapy or integrative medicine.

Conclusion:

Summarizes key findings concisely. Reinforces the relevance of the four-step model.

Figures and Tables:

Well-structured and visually informative. Clearly labeled and appropriately referenced in the text. Effectively illustrate the proposed framework.

General statement: minor revisions

While the manuscript presents an innovative and promising conceptual framework for osteopathic practice, several critical issues need to be addressed before publication in this journal as mentioned above. By addressing these issues, the manuscript will be significantly improved and more suitable for publication.

Author Response

Thank you very much for taking the time to review this manuscript. Please find below the detailed responses to your comments, along with the corresponding revisions and corrections, which are highlighted in red in the re-submitted files. The revised manuscript now has a word count of 8,000 words, three figures, and four tables. It also includes Appendix 1 and 2, with a total of 1,432 words. The Supplementary Materials consist of "Table S1" (4,583 words), which outlines the eligibility process by detailing the included articles, relevant content, related emergent themes, and additional references. Furthermore, a 5-minute video titled "Video S1: Patient-practitioner-environment synchronization: A four-step practical framework" has been added to enhance readers' understanding of the clinical applications of the proposed model. Additionally, a graphical abstract is provided. Below is a point-by-point answer to the reviewers' comments.

Reviewer 2. Point-by-point response to Comments and Suggestions for Authors

Comments 1: This manuscript explores the concept of patient-practitioner-environment synchronization in osteopathic care through a four-step process that integrates interprofessional collaboration and distinctive osteopathic competencies. The study aims to provide a theoretical and practical framework for osteopathic practice, informed by a scoping review of existing literature. The authors propose a model that combines narrative-based decision-making, touch-based shared sense-making, mindfulness-based osteopathic manipulative treatment (OMT), and patient-active participation to enhance bio-behavioral synchrony.

Response 1: Thank you for your insightful summary and constructive feedback on our manuscript. We appreciate your recognition of our approach to patient-practitioner-environment synchronization in osteopathic care and our effort to integrate interprofessional collaboration with distinctive osteopathic competencies. Your comments reinforce the relevance of our proposed framework, which combines narrative-based decision-making, touch-based shared sense-making, mindfulness-based OMT, and patient-active participation to enhance bio-behavioral synchrony. We are grateful for your thoughtful assessment and look forward to any further suggestions that may help refine our work.

Comments 2: Abstract: Clearly articulates the purpose and scope of the study. Provides a well-structured summary of the proposed framework.

Response 2: Thank you for your positive feedback on the abstract. We appreciate your recognition of its clarity and structure in summarizing the purpose, scope, and proposed framework of our study. Your comments reinforce our efforts to present a concise and comprehensive overview of our work. We remain available for any further suggestions for improvement.

Comments 3: Introduction:

Provides a strong rationale for reconceptualizing osteopathic principles. Engages with contemporary debates on tradition versus modernization in osteopathy. Positions the study within the broader context of interprofessional healthcare.

However, some sections are overly detailed, making the introduction lengthy. The transition to the proposed four-step framework is not well signposted.

Response 3: Thank you for your valuable feedback. We appreciate your acknowledgment of the strong rationale provided in the introduction and its engagement with contemporary debates in osteopathy. Regarding the length and level of detail, we recognize the importance of maintaining a concise introduction while ensuring clarity in positioning the study. We have refined the section to enhance readability without compromising essential background information. Concerning the transition to the proposed four-step framework, we deliberately refrained from introducing the framework in the introduction, as scoping reviews require a clear distinction between background information and results. Since the framework is derived from the literature review process, presenting it in the introduction could misrepresent the rigor and transparency of the review. Instead, we have included a more structured presentation of the three core aspects of osteopathic care that underpin the practical framework (lines 140-144):

"The authors aim to incorporate three core aspects of osteopathic care (OC) to structure the practical framework: (1) the application of the osteopathic palpatory diagnosis process (OPDP) and the identification of somatic dysfunction (SD), (2) a holistic approach grounded in osteopathic principles, and (3) the individualization of OC tailored to the unique needs of each patient."

This addition ensures that the conceptual foundation of the framework is clear while maintaining the methodological integrity of the scoping review. We hope these revisions address your concerns and enhance the clarity of the manuscript.

Comment 4: Materials and Methods:

Follows established guidelines for scoping reviews. Provides a transparent search strategy, including database use and keywords. Describes the process of thematic and qualitative analysis.

However, the selection criteria for articles are not clearly justified. The rationale for including snowball-sampled studies is not explicitly stated. No discussion of potential bias in study selection.

Response 4: We would like to thank the reviewer for their insightful feedback. In response to the comment regarding the selection criteria for articles, the rationale for including snowball-sampled studies, and the potential bias in study selection, we have made the following revisions to Section 2.3. Eligibility Criteria (Lines 201-216):

  1. Selection Criteria Justification: We have provided a more detailed and transparent explanation of the inclusion and exclusion criteria used in the selection of articles for this scoping review. As outlined in the Eligibility Criteria section, papers were included based on their alignment with three core aspects of osteopathic care (OC):
    - The application of the osteopathic palpatory diagnosis process (OPDP) and the identification of somatic dysfunction (SD),
    -A holistic approach rooted in osteopathic principles, and
    - The individualization of osteopathic care tailored to the unique needs of each patient.

  2. Articles that did not meet these core aspects, or those that focused on manipulative techniques or psycho-corporeal approaches without incorporating osteopathic clinical reasoning, were excluded from the review. The eligibility of papers was determined through a two-step process, with independent evaluations conducted by the expert team (G.D.’A., C.L., F.B.).

  3. Rationale for Snowball-Sampled Studies: We have clarified the rationale for including snowball-sampled studies. Given the emerging nature of this research topic, relevant articles were sometimes identified by reviewing the reference lists of selected studies. This method allowed us to capture additional relevant research that may not have been included in the primary search. We have explicitly stated that this approach was employed to enhance the comprehensiveness of the review and ensure that important studies that were not captured in the initial search were included.

Moreover, we implemented the discussion of Potential Bias in Study Selection: We have added a discussion regarding the potential biases in the study selection process. While PubMed (MEDLINE) was used as the primary database, we acknowledge that relying on a single database and the use of snowball sampling could have introduced selection bias. We discuss this limitation in the Limitations and Future Directions section (lines 692-703), noting that, while these strategies provided a comprehensive view of the literature, they may have excluded studies not indexed in PubMed or those not cited by the included studies.
These revisions improve the methodological clarity of the review and address the concerns raised. We believe these changes ensure a more transparent and robust explanation of the selection process, and we remain open to any further suggestions or requests for clarification. Thank you again for your constructive feedback.

Comment 5: Results:

Presents findings in a well-organized manner. Groups results by the four-step framework, making interpretation easier. Provides tables and figures to summarize key themes. However, no mention of how conflicting findings were handled. Tables could include more quantitative indicators to support thematic analysis.

Response 5: We appreciate the reviewer’s valuable feedback. In response to the comment regarding the suggestion to include more quantitative indicators to support the thematic analysis, we have revised the manuscript accordingly. Specifically, we have added Table 3, which organizes the results by the four-step framework, enhancing clarity and interpretation. This table also incorporates relevant quantitative data that complements the thematic findings, aiming to provide a more comprehensive and clearer understanding of the results. We believe these revisions address the reviewer’s concern and enhance the manuscript's overall quality.

Regarding the handling of conflicting findings, we have included in the discussion section how conflicting findings were addressed (lines 331-343). Specifically, we state: "The present scoping review proposes a practical framework to address the seemingly conflicting results arising from the ongoing debate surrounding the reconceptualization of osteopathic theoretical and clinical practice models. On one side, there are 'tradition-dismissive' authors [2-6], while on the other, 'tradition-reconceptualization' osteopathic practitioners (OPs) [7-22]. To address these conflicting findings, we identified the sources of discrepancy, discussed their implications, and offered possible explanations for the differences. We acknowledged these discrepancies and explored potential points of convergence through an enactivist perspective. While this review may not resolve the conflict, it transparently addresses these variations, transforming the debate into a potential dialogue on the evolving perspectives in contemporary osteopathic care. By following integrative hypotheses, we propose a practical framework for osteopathic approaches to bio-behavioral synchrony. Finally, we offer suggestions for future research to help resolve contradictions in the existing literature."

We trust these changes have addressed the concerns raised and further strengthened the manuscript. Thank you again for your insightful comments.

Comment 6: Discussion:

Effectively contextualizes findings within osteopathic and interprofessional healthcare literature. Proposes a novel framework that integrates tradition with modern practice. Highlights potential clinical applications.

However, does not sufficiently address the limitations of the review. Lacks direct comparisons with similar models in manual therapy or integrative medicine.

Response 6: We appreciate the reviewer’s insightful feedback.  We have included in the discussion the subsection titled "4.1.6. Integrating Contemporary Knowledge and Models: Positioning Osteopathic Care within the Broader Context of Manual Therapy and Integrative Medicine.(lines 731-803)" This subsection directly addresses the need for comparisons with similar models in manual therapy and integrative medicine by discussing the integration of contemporary knowledge, such as enactivism, within osteopathic care. We acknowledge the importance of building a comprehensive strategy that reconciles divergent views within the osteopathic community and aligns osteopathic care with broader healthcare models. This section emphasizes the need for contemporary osteopathy (OC) to preserve its unique identity while incorporating modern conceptual models that are relevant to manual therapy and integrative medicine. Additionally, it explores how OC can draw upon both manipulative and educational strategies to engage patients in health promotion, positioning osteopathic care within a wider interdisciplinary context. By recognizing the interconnection between osteopathy and other healthcare models, we aim to illustrate the broader applicability and relevance of osteopathic principles in modern healthcare practice.

In response to the comment regarding the need for a more thorough discussion of the review’s limitations, we have addressed this concern by incorporating a new paragraph in the discussion section, specifically in subsection 4.1.6. Limitations and future directions. (804-850) In this section, we acknowledge that one limitation of the scoping review is the reliance on a single database, PubMed (MEDLINE), for the literature search. While PubMed is a comprehensive and highly relevant resource for biomedical and healthcare-related literature, including osteopathy, the use of only one database may have constrained the breadth of the review. To mitigate this, we manually reviewed reference lists and employed a snowball sampling approach to identify additional relevant studies. Future reviews could benefit from incorporating multiple databases to enhance the comprehensiveness of the search. Additionally, we highlight a second limitation: the emerging nature of the research topic, which has resulted in a higher degree of self-citation. Given the limited availability of literature directly addressing the distinctive roles and contributions of osteopathic care (OC), the review includes studies authored by the same research team or those closely associated with their work. While self-citation is common in nascent fields, this could affect the perceived novelty of the findings. This challenge is further compounded by the lack of recent publications focusing specifically on OC’s unique contributions. Historically, much of the discourse has emphasized OC’s limitations rather than its strengths and developmental opportunities. We emphasize the importance of future research incorporating diverse external perspectives to further strengthen the field. We hope that this addition sufficiently addresses the reviewer’s concerns by transparently discussing the limitations of the study and suggesting directions for future research.

Comment 7: Conclusion:

Summarizes key findings concisely. Reinforces the relevance of the four-step model.

Response 7: We appreciate the reviewer’s positive feedback on the conclusion section. We are pleased that the summary effectively reinforces the relevance of the four-step model and concisely presents the key findings.

Comment 8:

Figures and Tables:

Well-structured and visually informative. Clearly labeled and appropriately referenced in the text. Effectively illustrate the proposed framework.

Response 8: We sincerely appreciate the reviewer’s positive feedback on the figures and tables. In response to the suggestion for additional quantitative indicators and clearer visualization, we have added Table 3 and Figure 3 to further support the thematic analysis and enhance the clarity of the proposed framework. We are pleased to hear that the figures and tables are well-structured, visually informative, and effectively illustrate the framework, as ensuring clarity and coherence in our visual representations was a key priority.

Comment 9: General statement: minor revisions

While the manuscript presents an innovative and promising conceptual framework for osteopathic practice, several critical issues need to be addressed before publication in this journal, as mentioned above. By addressing these issues, the manuscript will be significantly improved and more suitable for publication.

Response 9: We sincerely thank the reviewer for their constructive feedback and for recognizing the innovative and promising nature of the conceptual framework presented in the manuscript. We appreciate the opportunity to address the critical issues raised. We have carefully considered each of the points and have made the necessary revisions to enhance the manuscript’s clarity, rigor, and suitability for publication. We believe these changes will significantly improve the manuscript, and we are grateful for your valuable suggestions in helping us refine the work. If you have any further concerns or recommendations, we remain open to your insights.
